# Porcine Germ Cells Phenotype during Embryonic and Adult Development

**DOI:** 10.3390/ani13152520

**Published:** 2023-08-04

**Authors:** Amanda Soares Jorge, Kaiana Recchia, Mayra Hirakawa Glória, Aline Fernanda de Souza, Laís Vicari de Figueirêdo Pessôa, Paulo Fantinato Neto, Daniele dos Santos Martins, André Furugen Cesar de Andrade, Simone Maria Massami Kitamura Martins, Fabiana Fernandes Bressan, Naira Caroline Godoy Pieri

**Affiliations:** 1Department of Veterinary Medicine, School of Animal Sciences and Food Engineering, University of Sao Paulo, Pirassununga 13635-900, SP, Brazil; amanda_soares_jorge@usp.br (A.S.J.); may.h.gloria@gmail.com (M.H.G.); laisvpessoa@usp.br (L.V.d.F.P.); fantinato@usp.br (P.F.N.); daniele@usp.br (D.d.S.M.); fabianabressan@usp.br (F.F.B.); 2Department of Surgery, Faculty of Veterinary Medicine and Animal Sciences, University of Sao Paulo, São Paulo 01001-010, SP, Brazil; kaiana.recchia@usp.br; 3Department Biomedical Science, Ontario Veterinary College (OVC), University of Guelph, Guelph, ON N1G 2W1, Canada; alinsouza25@gmail.com; 4Department of Animal Reproduction, Faculty of Veterinary Medicine and Animal Sciences, University of Sao Paulo, Pirassununga 13635-900, SP, Brazil; andrefc@usp.br; 5Department of Animal Production, Faculty of Veterinary Medicine and Animal Sciences, University of São Paulo, Pirassununga 13635-900, SP, Brazil; smmkm@usp.br

**Keywords:** PGCs, reproduction, porcine

## Abstract

**Simple Summary:**

The lineage of germ cells (GCs) ensures the perpetuation and diversification of genetic information across generations in most vertebral organisms. As part of their development into functional gametes, PGCs undergo extensive genetic and epigenetic modifications during their specification and development. We aimed to understand the dynamics of pluripotent, germline, and epigenetic markers of porcine PGCs (pPGCs) during the gestational and adult periods in both genders. The results demonstrated morphological differences between embryonic/fetal ages and genders in the expression of pluripotency and germinal markers. Additionally, different patterns of epigenetic markers were found at these embryonic/fetal ages. These findings are significant to the field of germ cells because they exhibit the progression of a variety of markers during pPGCs, and they may allow their use in applied science such as reproductive and regenerative medicine.

**Abstract:**

Primordial germ cells (PGCs) are the precursors of gametes. Due to their importance for the formation and reproduction of an organism, understanding the mechanisms and pathways of PGCs and the differences between males and females is essential. However, there is little research in domestic animals, e.g., swine, regarding the epigenetic and pluripotency profiles of PGCs during development. This study analyzed the expression of epigenetic and various pluripotent and germline markers associated with the development and differentiation of PGCs in porcine (pPGCs), aiming to understand the different gene expression profiles between the genders. The analysis of gonads at different gestational periods (from 24 to 35 days post fertilization (dpf) and in adults) was evaluated by immunofluorescence and RT-qPCR and showed phenotypic differences between the gonads of male and female embryos. In addition, the pPGCs were positive for OCT4 and VASA; some cells were H3k27me3 positive in male embryos and adult testes. In adults, the cells of the testes were positive for germline markers, as confirmed by gene expression analysis. The results may contribute to understanding the pPGC pathways during reproductive development, while also contributing to the knowledge needed to generate mature gametes in vitro.

## 1. Introduction

For more than a century, the patterns and mechanisms that control the successful migration of germ cells have been a field of interest in reproductive biology [1,2]. Mammalian studies have shown that PGCs originate outside the gonads, in the proximal region of the epiblast, and migrate through the dorsal mesentery to the gonadal ridge during early embryonic development [3]. In humans, these cells originate during the first 2–3 weeks of embryonic development [4]. In mice, at an approximate embryonic age of 7.2 days (E7.2), somatic signals demarcate a group of nearby epiblast cells as potential precursors of germ cells [5,6].

In porcine embryos, the first clusters of PGCs were identified 11 to 12 days after fertilization. According to Kobayashi et al. [4], from E12 to E13.5, these PGCs exhibit the co-expression of SOX17, BLIMP1, NANOG, TFAP2C, and OCT4 markers and are formed by a group of 60 cells, with localization on the border between the embryonic and extraembryonic tissue. From E15.5, these cells proliferate, forming a group of approximately 300 PGCs [4]. Then, PGCs migrate from the primitive mesentery to the gonadal ridge. Between 1000 and 2000 PGCs colonize the gonads, and they undergo mitotic blockage. During this period, PGCs are morphologically large cells and can be identified using alkaline phosphatase staining, or by the expression of the OCT4 transcription factor, for example [3]. 

Several molecular mechanisms and signaling pathways are active during mammalian PGC specification [7,8]. In mice, BMP4 induces BLIMP1 and PRDM14 expression, along with AP2γ and TFAP2C, leading to PGCs fate. Other markers such as VASA, DND, OCT4, NANOG, SOX2, BLIMP1, DAZL, STELLA, and PIWI are expressed [9,10]. SOX17 and BLIMP1 are reported to be necessary for inducing human PGCs [11]. TFAP2C, PRDM14, and NODAL signaling play crucial roles in human germ cell specification, inducing PGC fate and suppressing the somatic program [11,12]. In humans, SOX17 is a key regulator of hPGCLCs (human primordial germ cell-like cells), and BLIMP1 represses endodermal and somatic genes. TFAP2C functions upstream of SOX17 and activates its expression. In adulthood, despite differences between genders, both STELLA and VASA were found during the development of germ cells [13,14,15].

The PGC cells undergo numerous processes from their emergence to their arrival at the genital ridge and can originate new individuals; hence, germ cells represent a unique cellular type. However, in some mammalian species and humans, when and how germ cells acquire this resource in embryogenesis, or how they carry out the migration process are essential questions in biological development that remain unanswered [16]. In mammals, PGCs are reported to undergo extensive epigenetic reprogramming during embryonic development, including an almost complete demethylation of the DNA. These findings were more fully explored in mice, and further investigations should be carried out in other species [17,18,19].

It is known that the genetic, transcriptional, and epigenetic profiles of PGCs in humans and mice have some different characteristics, which may be related to differences in pluripotency status or even post-implantation development [20]. In this sense, new models of studies that can provide information on the development of germ cells are important.

The porcine, in this context, is a viable and highly advantageous model in regenerative studies and in other fields of reproduction, as it is considered a translational model that presents anatomy, physiology, and metabolism close to humans. The results are highly applicable and can be used in research in human and veterinary medicine, such as xenotransplantation [21,22,23].

However, knowledge about the in vivo and in vitro profiles of PGCs in porcine is still scarce [4,17,24,25,26,27]. Some authors have shown that these cells exhibit different levels of epigenetic changes during migration until reaching the genital ridge [28,29]. In a more recent study, Kobayashi et al. [4] showed the emergence of pPGCs and the expression of several important markers for the specification of these cells during the early development period. There is still little information about the mechanisms and pathways in which these cells are involved.

Therefore, this study aimed to determine the phenotypic profiles of porcine PGCs during embryonic development and in adulthood and evaluate the dynamics of germinal and pluripotency markers of these cells through immunofluorescence and RT-qPCR techniques. The knowledge acquired in these studies is expected to increase our understanding of the initial mechanisms of reprogramming germ cells during the reproductive process. Furthermore, we aim to understand the dynamics of epigenetic changes in these cells and the differences between such events in females and males. The generation of new knowledge about porcine PGCs in vivo can be transferred to the generation of mature gametes in vitro, which can, in turn, be used both in reproduction biotechnologies in domestic animals and as translational models [30,31,32]. 

## 2. Materials and Methods

### 2.1. Sample Collection

All procedures performed in this study were conducted in accordance with the Ethics Committee for the Use of Animals of the Faculty of Veterinary Medicine and Animal Sciences, University of São Paulo (CEUA 1239200616; CEUA 5130110517) and the Faculty of Animal Science and Food Engineering (CEUA 8046210520; CEUA 3526250717). The embryos and fetuses used in this experiment were collected from pregnant domestic porcine (*Sus scrofa*) females of a commercial hybrid strain (DB-90, DB Genética Suína, Patos de Minas-MG) at the Swine Research Center, School of Veterinary Medicine and Animal Science, University of Sao Paulo, Brazil. Females (n = 5) were submitted to artificial insemination, and embryos (n = 3 per age and sex) from 24 to 35 days dpf were used in this experiment. For all experiments, only females with three or more estrous cycles, aged approximately 220 days, and a body weight of ±130 kg were used. The animals were fed twice a day with commercial Agroceres Multimix^®^ pelleted feed, respecting the requirements of the NRC (1998). Access to water was ad libitum provided by pacifier-type drinkers.

The testes (n = 3) and ovaries (n = 3) of adult porcine were obtained from the slaughterhouse of the Faculty of Animal Science and Food Engineering, University of São Paulo, located on the Campus of USP “Fernando Costa”, Pirassununga.

### 2.2. Sexing of Collected Embryos

The embryonic tissues were collected, and DNA was extracted as previously described [33]. PCR analysis was performed using specific primers for the *SRY* gene (forward sequence 5′-TTCTGCAACAGGAGGATCGC-3′ and reverse sequence 5′- CACGGT GAAAAGGCAAGTCG—NM_214452.3) and the endogenous *Zfx* (female)/Zfy (male) [34,35]. PCR reactions had an initial denaturation step of 1 min at 95 °C, followed by 40 cycles of denaturation at 95 °C for 30 s, reheating at 60 °C for 1 min and elongation at 72 °C for 1 min, followed by final elongation at 72 °C for 1 min. The testicles and ovaries of adult porcines were used as positive controls. PCR products were analyzed using 2% agarose gel.

### 2.3. Histological Analysis

Embryos were collected, analyzed regarding the development of morphogenesis, and measured using the Crown-Rump (CR) technique [36,37,38]. For the morphological analysis, the samples were dissected and fixed in 4% paraformaldehyde for 24 h, and the microscopic characteristics were identified. The slides with samples were submitted to routine histological techniques. Briefly, the samples were dehydrated with a series of alcohols in increasing concentrations (70, 80, 95, and 100%), diaphanized in xylene, and embedded in paraffin [36]. For analysis, sections of 3 μm were cut and stained with hematoxylin-eosin (HE). All data were processed using a Zeiss microscope Axioplan 2 Imaging (Jena, Germany) and the images were captured at 20× and 40× magnifications with Zeiss Zen software (ZEN 2.6, blue edition).

### 2.4. Immunofluorescence Analysis

For immunofluorescence analysis, the samples were processed as previously described [37,38]. The samples were fixed for 24 h in 4% paraformaldehyde solution, dehydrated, embedded in paraffin, and sectioned with a thickness of 3µm. The slides were deparaffinized in xylene and hydrated in a series of 100% (*v*:*v*) to 70% (*v*:*v*) ethanol, followed by exposure to 0.5% Triton-X 100 in PBS (buffered saline) for 10 min at room temperature. Then, the slides were microwaved in a 0.01 M citrus reagent pH 6.0 solution for antigen recovery for 15 min at 92 °C. Then, the slides were washed in PBS-T (0.5% Triton buffered saline) at pH 7.2. The slides were blocked with 1% bovine serum (BSA, Sigma-Aldrich Corp., St. Louis, MO, USA) and 0.1% Tween-20 (PBS-T) for 30 min. Primary antibodies: OCT4 (POU5F1, 1:100 cat# sc8628), NANOG (1:100, cat#ab77095), STELLA (Dapp3, 1:100, cat#sc67249), VASA (DDX4, 1:200, cat# ab13840), c-Kit (1:50, cat# sc1494), STRA8 (1:100, cat#ab49602), DAZL (1:100, cat# sc27333), and PLZF (ZBTB16, 1:50, cat#Sc22839), were diluted in blocking solution and incubated at 4 °C overnight. To analyze epigenetic markers, the samples were permeabilized with 0.5% Triton-X 100 in PBS for 20 min at room temperature. The slides were heated in the microwave in a 0.01 M citrus reagent pH 6.0 solution with 0.05% tween 20 for 10 min at 92 °C. Then, samples were washed in PBS-T (buffered saline solution with 0.25% Triton) at pH 7.2. The slides were blocked with 1% bovine serum (BSA, Sigma-Aldrich Corp., St. Louis, MO, USA) and 0.05% Tween-20 (PBS-T) for 30 min at 37 °C. The primary antibodies, H3k9me2 and H3k27me3 (1:250; 7449/ Millipore; 7441/ Millipore), were diluted in a blocking solution and incubated at 4º C overnight. After incubation, samples were washed in PBS-T (0.5% Triton-buffered saline) at pH 7.2 and incubated with the secondary antibody for 1 h at room temperature. Cell nuclei were stained with Hoechst (Trihydrochloride, Trihydrate, cat#33342, Invitrogen, Carlsbad, CA, USA). Finally, slides were mounted with ProLong Gold (Life Technologies cat# P36930, Life technology; Carlsbad, CA, USA) and stored at 4 °C. Negative controls were performed in the absence of primary antibodies. All data were processed using the Zeiss microscope Axioplan 2 Imaging, and the images were captured at 20× and 40× magnification with Zeiss Zen software (ZEN 2.6, blue edition).

### 2.5. qRT-PCR 

All RNA extractions prior to RT-qPCR analysis were performed using Trizol Reagent (Invitrogen) and quantified by the Scientific NanoDrop™ spectrophotometer. The relative abundance of gene transcripts was determined using RT-qPCR analysis based on the protocol by [39]. All primer sequences were designed using the Primer designing tool NCBI–NIH [4,26,27] (Table 1). All reactions were performed in biological triplicates and technical duplicates using *GAPDH* and *β-ACTIN* expression as housekeeping reference genes. A pool of porcine blastocysts was used as a positive control and adult fibroblasts as a negative control. Data are artificially provided by Step One Software 2.3 and analyzed with LinRegPCR software (version 2015.0).

### 2.6. Statistical Analysis

The data obtained from the experimental procedures were analyzed using the SAS University Edition statistical program. Previous verification of the normality of the residues was carried out using the Shapiro–Wilk test. Variables that did not meet the statistical assumptions were submitted to a logarithmic transformation [Log (X + 1)]. The original or transformed data, when this procedure was necessary, were submitted to the Analysis of Variance and Tukey’s test for comparison between the means of the different experimental groups. In all statistical analyses, the significance level considered was 5% (*p* < 0.05).

## 3. Results

### 3.1. Morphological and Immunofluorescence Analysis 

#### 3.1.1. Male Gonads 

In the morphological analysis of male embryos at 24 dpf, an elongated and thickened region can be seen on the medial wall of the mesonephric duct, originating from a small protuberance in the body cavity. In addition, it was possible to detect a few large cells with rounded and prominent nucleoli (Figure 1A,B). There is no clear tissue organization, just as there is no well-defined border. In this period, the gonads present large, rounded cells with a prominent nucleolus and co-localization with OCT4 and VASA, in addition to some cells close to the STRA8-positive basement membrane (Figure 1C). At 25 days of gestation, the thickening of the medial wall of the mesonephric duct was observed, where a small prominence originates towards the body cavity (Figure 1A,B). The gonads present large, rounded cells with a prominent nucleolus and co-localization of OCT4 (red) and VASA (green) (Figure 1C). At 29 days, we observed the mesonephros in prominent tubules located laterally of the future urogenital system and laterally to the gonadal ridges; however, they are still undifferentiated at this stage (Figure 1A,B). The PGCs were co-localized for OCT4 (red) and VASA (green) (Figure 1C).

At the initial period (24–29 dpf), it was possible to visualize the gonadal ridges and, laterally to it, the mesonephros. The tubules of the mesonephros are composed of epithelial cells, which form a simple cubic epithelium and are wrapped around blood vessels and a protective serous layer. In the caudal portion of this organ, it is possible to detect the presence of the metanephros tissue, or primitive kidney (Figure 1B). 

At 35 days, the gonads separate from the mesonephros, and it is already possible to observe the presence of testicular cords composed of centrally located primordial germ cells and presumably peripherally located supporting cells, or Sertoli cells. Germ cells between 30 and 35 dpf were positive for VASA (green) and the co-localization of OCT4 (red) (Figure 1B,C). 

Histological analysis of the porcine newborn testes with from 0 to 3 days of life showed some large and rounded cells in the center of the seminiferous tubules that presented morphological characteristics of undifferentiated spermatogonia and some Sertoli cells that were located closer to the basement membrane (Figure 2A). At this age, spermatogonia were positive for PLZF, STRA8, VASA, and STELLA (green) and negative for C-kit, DAZL, NANOG, and OCT4 (red) (Figure 2B).

The morphological analysis of the testicles of adult males (Figure 3A) showed the testicle surrounded by the tunica albuginea, from which the fibrous septa that divide the testicular parenchyma into testicular lobules are formed by seminiferous tubules surrounded by loose connective tissue rich in blood vessels and interstitial cells (Leydig cells, testicular androgen-secreting cells). The seminiferous tubules were formed by the tunica propria, seminiferous epithelium, and tubular lumen. In the seminiferous epithelium, undifferentiated spermatogonia and Sertoli cells were found close to the basement membrane. The lumen was composed of spermatocytes, spermatids, and spermatozoa (Figure 3(A1–A3)). During this period, some rare undifferentiated spermatogonia located close to the basement membrane were PLZF (green) and OCT4 (red) positive (Figure 3B).

The differentiated spermatogonia and primary spermatocytes were VASA-positive (green). DAZL (red) and STRA8 (green) were detected in differentiated spermatogonia close to the tubules’ basal membrane and primary spermatocytes, and some spermatids were STRA8-positive (green). Co-immunolocalization of STRA8 and DAZL and some primary spermatocytes. Furthermore, spermatocytes were c-Kit-positive (red) (Figure 3B). However, the cells were negative for NANOG (red) and STELLA (green).

Embryos at 24, 29, and 35 dpf in the neonates tests were negative for histone alteration markers (H3k9me2 and H3k27me3) detection, although H3k27me3 was detected in some cells at 25/26 days of gestation and in spermatocytes found in adult porcine testes (Figure 1B, Figure 2B and Figure 3B). The dynamics of global methylation (5 mC) and cytosine hydroxymethylation (5 hmC) at 25 and 29 days of gestation showed positive OCT4 (red) and 5 mC cells (green) (Figure 1C). However, after birth, the cells were negative for both markers in the testes evaluated in this study.

#### 3.1.2. Female Gonads 

The PGCs of the female porcine gonads had the following morphological characteristics: large, rounded cells with prominent nucleoli, as observed in germ cells of other species (Figure 4A). Embryos at 24–29 days showed similar morphological features to the male porcine embryos. At 29 dpf, many OCT4-positive cells (red) were detected, and some cells were VASA, STELLA, and STRA8-positive (green) (Figure 4B). At 35 days, the gonads showed morphological differences, but it was not possible to detect OCT4-positive cells (Figure 4B). However, the primitive gonads were practically independent of the mesonephros and had a different cellular arrangement than the previously analyzed ages.

In the morphological analysis of the ovaries of adult female porcine at reproductive age (Figure 5A), the complete oogenesis process was observed, and follicles at different stages of maturation were identified, such as primordial follicles composed of oocytes surrounded by a single layer of flattened follicular cells; uni- or multilaminar primary follicles with the beginning of zona pellucida formation; secondary or preantral follicles with formed zona pellucida, a layer of granulosa cells and theca interna; tertiary or antral follicle formation of the antrum, theca organized into internal and external and organization of granulosa cells; Antrum formed, thick theca layers and oocyte surrounded by corona radiata and *cumulus oophorus* cells in Graafian or pre-ovulatory follicles (Figure 5(A1–A5)). Immunofluorescence of germ cells showed cells positive for NANOG (red), VASA (green), STELLA (green), STRA8 (green), and DAZL (red) (Figure 5B) detection.

Analysis of the histone alteration markers H3k27me3 and H3k9me3 showed cells negative in the periods of 25/26, 29-, and 35 dpf, although in the adult ovaries, follicular cells were positive for the detection of H3k9me2 (green), in the primary follicle, and H3k27me3 (green), in interstitial cells, and in the cells that comprise the theca of Graff’s follicle (Figure 5B). 

### 3.2. Analysis of the Gene Expression in Porcine Germ Cells

Gene transcription analysis showed the transcript abundance of pluripotent genes (*OCT4 (POU5F1), SOX2, NANOG*) and germline genes (*PRDM14, BLIMP1 (PRMD1*), *SOX17, TFAP2C, VASA (DDX4), STELLA (Dppa3), DAZL,* and *STRA8*) regarding porcine germ cells from both genders at four different periods, and in most of them, also in neonatal testes and adult ovaries or testes. When germ cells were analyzed at different periods, the expression of *OCT4, PRDM14,* and *DAZL* was increased in 35 dpf, regardless of sex (*p* < 0.05, Figure 1D). Then, the germ cell expression profile was compared to the adult tissues, and differences were observed for *SOX2, NANOG, STELLA, VASA,* and *DAZL* (*p* < 0.05, Figure 2C). *SOX2, STELLA,* and *DAZL* presented a higher transcript abundance in adult testicles when compared to embryonic stages. *NANOG* presented a higher expression in the ovaries (*p* < 0.05, Figure 2C and Figure 3C). 

Specifically in males, *SOX2* was increased in adult testicles, and *VASA* and PRDM14 increased in neonatal testes (adult tissues were not studied regarding *PRDM14*). *STELLA, DAZL,* and *VASA* expression was observed and visually increased in adult testicles; however, it was not statistically different, possibly due to the high heterogeneity between groups. *SOX17, TFAP2C,* and *PRDM14* were observed during embryonic periods, and *TFAP2C* and *PRDM14* showed similar patterns (Figure 3C).

The *OCT4* expression was not observed in ovaries; however, it was more abundant in 35d females (*p* = 0.0640). *SOX2* was visually decreased in females as age increased, as expected. *NANOG* is a gene that exists in the germline of females at different gestational and adult periods (*p* < 0.0175, Figure 2C and Figure 5C). *NANOG, STELLA, VASA DAZL,* and *STRA8* were higher in adult females; however, they were not statistically different, probably due to a high standard deviation. *PRDM14* presented a higher abundance at 35 dpf, and *BLIMP1* decreased its expression as the age increased with the development of the gonads (Figure 5C). 

When comparing the relative gene transcriptional abundance between genders (Figure 6), interestingly, it was possible to detect a variation in gene *SOX17* expression, which is important for specifying PGCs (*p* = 0.0076), when gender was analyzed regardless of gestational age, which increased in females. The *TFAP2C* expression profile, however, was not different and was visually higher in males. Additionally, as previously described, *OCT4* was increased at 35 dpf in both genders, *SOX2* increased in adults, particularly in males, and *DAZL* was higher in adults as well; however, other differences may not have been observed due to the high heterogeneity between groups.

## 4. Discussion

Previous animal studies have discussed germline markers’ expression during the PGC differentiation process [19,27,37,38,40], which can be influenced by age and sex. In this study, we described the phenotypic profile of germ cells in male and female porcine at different stages of embryonic development and adulthood through morphological analysis and of pluripotency and germinative markers associated with the development of porcine PGCs, performing a comparative analysis of the transcript abundance of these markers between males and females at different stages of reproductive development.

### 4.1. Histological and Immunophenotype Analysis of the Porcine Germ Cells

As observed in dogs [38], the primitive porcine gonads appear at the beginning of pregnancy, located laterally to the mesonephros. In mice, the genital ridge appears for the first time on the coelomic surface of the mesonephros at around 10 days post coitum (dpc), in humans at the fourth week of gestation (WG) [41], and in cattle at approximately 32 days of gestation [42]. In this period, the mesonephros has prominent tubules composed of epithelial and blood cells. In the most caudal region, the presence of metanephric tissue that will form the primitive kidney is observed, as observed in porcine embryos with 25 dpf. According to Parma et al. [24], in porcine embryos with 21 dpc, the gonads cannot be distinguished from mesonephros, and at 23 dpc, they showed similar microscopic characteristics in both genders. Only between 26 and 28 dpc was it possible to notice the morphological formation of the testes, with the formation of the testicular cords. In this study, the differentiation of the primitive gonads was observed in embryos after 30 dpf. At 35 days, it was already possible to notice morphological changes, making it possible to identify the different characteristics of each sex, marking the sexual differentiation, which was also reported by Lee et al. [43] In dogs, sexual differentiation occurs at 30 days of gestation, when PGCs begin the maturation process [38].

According to Hyttel et al. [3], in porcine embryos between 24 and 35 days, OCT4 expression is expected, regardless of sex, which is also confirmed by Hyldig et al. [17] and found in this study. In mice, OCT4 is detected in PGCs during the specification process, and these remain until the process of sexual differentiation occurs [44]. The OCT4, SOX2, and NANOG are essential in the process of the specification and proliferation of PGCs. Although PGCs are unipotent cells, they regain the expression of key genes for pluripotency after their specification, maintaining OCT4 expression and re-expressing SOX2 and NANOG [45]. Studies in mice have shown the importance of OCT4 in the maintenance and development of PGCs [46].

In porcine, the VASA transcript and protein were detected in embryos and adult germ cells, as described in humans, in both genders [15,47]. They described the expression of VASA in embryos from 24 to 61 days of gestation, ovary adult, oocytes, and testes. In humans, VASA is important for germ cells. It is expressed during gonadal ridge colonization and later in development, including gametogenesis [15]. Another study in mice showed the function of this gene in PGCs, when the male mice are homozygous for a targeted mutation of the mouse vasa homolog (Mvh); they are sterile, with apoptosis of the germ cells in the zygotene stage of meiosis and exhibit severe defects in spermatogenesis, whereas homozygous females are fertile [40].

In male embryos, in all periods analyzed (24–35 dpf), co-expression of the OCT4 and VASA proteins was detected, with the STRA8 protein being detected specifically in embryos of 24 days of gestation, and the transcripts of this gene in the same period and at 28 dpf, similar to that described by Pieri et al. [27] and Lee et al. [47], who have described the co-expression the protein VASA with OCT4 in PGCs at 28 days.

Pieri et al. [27] described the different levels of germline and pluripotent markers detected in male porcine embryos of different gestational ages, and, in other species, such as canine embryos, the VASA marker was detected after 27 days of gestation, and DAZL was present in embryos after 30 days of gestation [37] and in humans [40]. During embryonic development [25], the transcript of *DAZL* was observed in porcine embryos between 28 and 51 days of gestation, and the marker was present from the 31st day of gestation, in addition to the expression in testes of adult porcine, as well as observed in the gonads of female porcine at 35 days of gestation, and in the testes. *DAZL* (deleted in azoospermia-like), a germ cell-specific RNA-binding protein, is reported essential for the development of PGCs [48]. In mice, loss of *DAZL* expression results in infertility in both genders, with post migratory germ cell apoptosis and a final block in meiosis [49]. It is an important factor for PGCs initial to the meiosis, and *DAZL* is a necessary “licensing factor” for the sexual differentiation of PGCs [50,51].

In male neonates (1–3 days old), cells in the seminiferous tubules presented the markers VASA, DAZL, and STRA8, as evidenced in the gene transcription abundance analysis. Furthermore, PLZF (promyelocytic leukemia zinc finger, ZBTB16) protein was detected in spermatogonial cells. PLZF is a marker of undifferentiated spermatogonia conserved in large animals [52]; it plays an important role in developing, maintaining, and differentiating germline stem cells. Moreover, PLZF has been reported to be essential for testicular development and spermatogenesis in different species [53]. According to Goel et al. [54,55], this germline marker is found in newborn porcine, mainly in undifferentiated spermatogonial cells at 20 weeks of age. 

In mice, the 1-day-old neonatal testes are composed of testicular cords containing Sertoli cells; peripherally located and more centrally located gonocyte cells (pro-spermatogonia) have the PLZF, GFRA1, and CDH1 markers; and a small number of cells have the STRA8 cells or c-KIT markers. After 4 days of birth, it is already possible to identify two spermatogonial populations [56]. In porcine, PLZF and STRA8 are detected in seminiferous tubules, and qPCR analyses revealed the increased expression of VASA. The STRA8 (stimulated by retinoic acid gene 8) is expressed by germ cells in response to retinoic acid. In mice of both genders, retinoic acid (RA) produced in somatic cells acts directly on germ cells to induce Stra8 expression, which in turn is required for the initiation of the meiotic program. Thus, in females, it is expressed by embryonic ovarian germ cells shortly before they enter meiotic prophase [57]. In male mice, Stra8 is expressed postnatally in mitotically active cells of spermatogenic lineage (spermatogonia, pre-leptotene spermatocytes) [58,59]. In porcine, the STRA8 was evidenced in both genders; in females, the protein was detected at 30–35 dpf and in the ovaries (primordial follicles), and in males, it was detected in newborn and adult tests (spermatogonial cells and primary spermatocytes).

As expected, the porcine adults with more than 150 days of life presented complete spermatogenesis, with spermatogonia, spermatocytes, spermatids, and spermatozoa. We also detected the proteins DAZL, VASA, STRA8, PLZF, and c-Kit, as described in dogs and porcine [58,60,61,62]. Here, OCT4 and PLZF were detected in undifferentiated spermatogonia close to the basal membrane, as described in dog testicles [62]. The testes of porcine at different ages (7, 90, and 150 days) were analyzed by Zhang et al. [63] and the detection of a higher number of VASA+ cells after 150 days of age was reported. Another study described the detection of VASA in the testes of post-pubertal porcine (120 days) in spermatocytes but not in spermatogonia [14]. Furthermore, the authors mention the detection of c-Kit in spermatocytes and spermatids, as observed in this study in adult porcine. In mice, C-kit is expressed by migratory mPGCs, preleptotene spermatocytes, and type A, intermediate, and type B spermatogonia, but not in undifferentiated spermatogonia [64]. C-Kit is a transmembrane protein receptor associated with germ cell maturation, which marks the loss of spermatogonial stem cells potency and is expressed until the onset of meiosis [14]. 

In female embryos, at 25/26 dpf, cells positive for OCT4 and VASA were found; at 29 days, in addition to these markers, STELLA and STRA8 were also detected, while at 35 days, VASA was detected in some cells, similar to that described in female dog embryos at 35 days of gestation [37], The authors mentioned the detection of markers for DAZL and STELLA. In humans, OCT4, DAZL, and VASA were detected in fetal testes and ovaries at different gestational ages (14–20 weeks) [40].

In adult females of reproductive age and complete oogenesis, it was possible to histologically observe the follicles in their different stages of maturation, from primordial follicles to tertiary or Graafian follicles, which was expected according to what was described in other species [3]. A review described differences in the development pattern of follicles in different species of domestic animals; the porcine, unlike other species, have many follicles, with 30–90 follicles between 1 and 2 mm in diameter [65], similar to what was observed in this study. 

In the porcine ovaries, we observed the *DAZL* transcript. It was confirmed with the detection of this marker in primordial follicles and in the cytoplasm of oocytes, similar to that described in humans, where DAZL was detected in the cytoplasm of oogonia and follicular oocytes in fetal ovaries and adults. This marker, which plays an important role in the differentiation and maturation of germ cells [66,67], was detected in the granulosa of the primordial follicle [66,68,69]. Other germ cell markers were detected in porcine ovaries, including VASA and STELLA and the pluripotency marker NANOG; these were confirmed by gene transcript abundance analysis. The developmental pluripotency-associated protein 3 (Dppa3, also known as STELLA or PGC7) is a maternal gene expressed in germ cells, oocytes, and early embryo development [13]. In mice, failure in expression of STELLA leads to abnormal or deficient generation of oocytes [13,70,71]. In addition, some studies have shown that STELLA modulates the transcriptional program and regulates epigenetic modification, although it has a different role in maternal and paternal genomes [70,72].

### 4.2. Epigenetic Markers in Porcine Germ Cells

The pPGCs of both genders were evaluated at various gestational and adult ages, and H3k9me2 was not found. After puberty, H3k27me3+ cells were detected in the testes and ovaries; some of these markers show a progressive reduction in H3K9me2 expression, and global DNA demethylation ensues as pPGCs migrate toward the gonads [4]. Furthermore, pPGCs continued to express *SOX17*, *BLIMP1*, and *TFAP2C*, as observed in this study. 

The global metalation (5 mC) was detected in male embryos of 25–29 dpf. Intense demethylation occurs during germ cell development and at the beginning of embryonic development, compatible with the need to “redefine” the chromatin environment and restore the totipotency and pluripotency of the zygote and several stem cell populations. Such reprogramming steps involve the conversion of 5-methylcytosine (5 mC) to 5-hydroxymethylcytosine (5 hmC), a likely intermediate in DNA demethylation. In mice, during the beginning of the specification of PGCs, the levels of 5 mC are the same as in other somatic cells [73,74]. However, during migration and colonization, there is a reduction in global levels of DNA methylation, and 5 mC levels fall [75]. In males, demethylation occurs passively, while in females, there is a drastic loss of 5mC, called active demethylation [76]. Passive DNA demethylation depends on DNA replication and involves the reduction of overall 5 mC levels [77]. Active demethylation involves enzymatic reactions and the conversion of 5 mC into 5 mC by the action of an enzyme from the TET family [78]. Therefore, if considering that the pattern observed in mice is maintained in porcine, in all gestational and reproductive periods analyzed in male porcine, in all germ cells, 5 hmC was not detected. However, in embryos of 25/26–30 days of gestation, when the arrival of germ cells in the genital ridge and the differentiation of this organ occurs, the expression of 5 mC was observed. Studies in dogs have reported the detection of 5 mC and 5 hmC in the gonads of male canine embryos and fetuses at 25, 27, 28, and 30 days, and the presence of 5 mC was detected in the cells that make up the gonadal ridges [38]. Moreover, 5 hmC was not detected for all ages addressed by the author, both in embryos and neonates.

In mice, PGCs from 6.75 dpc embryos present active markers H3K4me2, H3K4me3, and H3K9ac, along with repressive markers H3K27me2 and H3K27me3, similar to those found in somatic cells. Thus, H3K27me3 is upregulated when PGCs start to migrate around 8.25 dpc; at 9.5 dpc, almost all PGCs show high levels of H3K27me3, and this remains elevated in gonadal mPGCs, suggesting an important role in maintaining genomic integrity during the period of active global demethylation [79,80]. In contrast, in porcine, pre-migratory pPGCs exhibit the onset of epigenetic reprogramming shortly after the specification phase, characterized by a global reduction in DNA methylation and H3k9me2 [4,17,28]. At the beginning of pPGCs migration (E17) and early gonadal (E25), the H3k27me3 marker is highly detected and markedly decreased in intermediate and late gonadal PGCs, as well as in humans [81]. After colonizing the gonads, pPGCs undergo asynchronous demethylation of imprinted genes, similar to human PGCs [81]. In the porcine embryos evaluated, no detection of H3k27me3 was observed in pPGCs from 25/26 dpf embryos, although the H3k9me2 marker was not detected throughout the evaluated gestational period. In vitro, according to Pieri et al. [27], pPGCLCs have both markers, H3k9me2 and H3k27me3, in a small population of cells, and 5 hmC was detected. Furthermore, the authors mention that in vivo, in male gonads with E26 and E29/30 days of gestation, it is possible to detect 5 mC positive cells. 

The H3K27me3 was detected in the testes of adult porcine, in spermatocytes and round spermatids, but not in elongated spermatids, similar to what was described by An et al. [82], who demonstrated the detection of H3K27me3 in spermatogonia, in intermediate spermatocytes, and in round spermatids, and the disappearance of this histone marker on the elongated spermatids. In mice and humans, H3K27me3 is present in mature sperm and is considered an important regulator in sperm development, regulating genes related to meiosis [83,84]. In humans, 12–30% of spermatozoa present heterogeneity in the distribution of H3k9me2, among others such as H3K4Me1, H3K4Me3, H3K79Me2, and H3K36Me3 [85], unlike in newborn or adult porcine, where we did not find this epigenetic marker; however, An et al. [82] mention that H3K4me2/3, H3K27me1/2/3, and H4K20me3 are found in the gonocytes of newborn porcine.

### 4.3. Analysis of the Gene Expression in Porcine Germ Cells

Herein, the transcripts’ abundance of pluripotency and germ cells related-genes and the development of PGCs and adult life was analyzed in porcine. pPGCs were found in ridge genitals at 24 dpf in male and 25/26 day in female embryos. Pluripotency and early specification genes were expressed in this stage, and *OCT4, PRDM14* and *DAZL* were increased after (35 dpf). In porcine, according to Kobayashi et al. [4], pPGCs are specified between days 12–14 of gestation after sequential regulation of *SOX17* and *BLIMP1* in response to BMP signaling. Then, the pPGCs begin the migration process at about E15 days through the hindgut and reach the gonadal ridges at 22 days of gestation, and undergo extensive proliferation between 28 and 42 days of gestation [29]. In mice, during the specification process, which takes place at 6.25 dpc, the bone morphogenetic protein (BMP) and WNT signals from extra-embryonic tissues induce of the *BLIMP1* gene, which represses somatic mesodermal genes in PGCs [86]. At 7.5 dpc, mPGCs positively regulate the genes related to this process, including *TFPA2C* and *PRDM14*. 

In embryos at E12.5-13.5, the porcine PGCs exhibit the co-expression of SOX17, BLIMP1, NANOG, OCT4, and TFAP2C proteins [4]. In this study, porcine embryos also presented the same markers in both genders, although they were evaluated by gene transcript abundance and in further gestational periods. Curiously, *SOX17* was increased in females regardless of age. A previous study in humans investigated the potential role of SOX17 involving germ cells maturation after specification, specifically in germ cells from a female fetus and adult gonads. The authors described that they detected SOX17 in both the cytoplasm and nucleus of oogonias at 7 gestational weeks (GW), and oocytes of primordial follicles, primary follicles from 15 to 28 GW, and in the nucleus of oocytes in secondary follicles, although the mechanism and SOX17 function in mature germ cells is unclear. Therefore, currently, there may not be enough information in the literature to form a conclusion about the difference in SOX17 expression between males and females [87]. 

Zhu et al. [81] analyzed the pPGCs of porcine embryos with 14 and 31 days of gestation, in the pre- and post-migratory phase, and they detected the transcripts of the genes *PRDM1 (BLIMP1)*, *TFAP2C*, *NANOS3*, and *c-Kit*, and the pluripotency genes *NANOG* and *OCT4* were increased, whereas *SOX2* was decreased. Similarly, we observed an upregulation of *OCT4* in the embryonic gonads at 35 days, regardless of gender; however, this was not found in NANOG or the SOX expression in this same embryonic period. The SOX2 (SRY-Box Transcription Factor 2) belongs to the family of HMG box transcription factors related to SRY, which are considered important in the maintenance of pluripotency in early embryonic cells and in the formation of various epithelial tissues during fetal development [88]. In the testicles, the expression of rare individual *SOX2 +* cells (spermatogonia A) was demonstrated in some seminiferous tubules, demonstrating a low transcriptional activity of this gene. The authors mention that *SOX2+* adult stem cells originate from fetal Sox2+ tissue progenitors [89]. 

According to Hyttel et al. [3], in porcine embryos between 24 and 35 days, a detectable transcript of *OCT4* is expected, regardless of gender. In mice, *OCT4* is detected in mPGCs during the specification process, and these remain until sexual differentiation occurs [45]. In female mice, a decrease in OCT4 expression in the adult ovary was observed, this may be related to a decrease in the number of PGCs that have potential pluripotency, and an increase in atretic and antral follicles [90,91], similarly as observed in porcine.

In porcine embryonic gonads, *VASA* and *STELLA* were present, but not increased at 35 dpf, and *STRA8* visually increased at 25–28 and 35dpf, as well as *DAZL* at 35 dpf, corroborating with our findings in immunofluorescence. During embryonic development, the *DAZL* transcript abundance in porcine embryos between 28 and 51 days of gestation was analyzed, and transcripts of this marker were detected from the 31st day of gestation, in addition to the transcript abundance in testes of adult porcine, as well as in the gonads of female porcine at 35 days of gestation. The *DAZL* gene belongs to the gene family of *DAZ* (Deleted in Azoospermia). The *DAZL* is conserved in vertebrates and required throughout germ cell development, including in germ line progenitors [92,93]. This gene has been shown to be expressed in the gonads of mice, humans and porcine, being necessary for the commitment and maintenance of the germ line [19].

Interestingly, when comparing male and female embryos, *DAZL*, *VASA*, *STELLA* and *SOX2* were increased in the adult phase in males, with a decrease in *STRA8* during this period, while in female embryos, *DAZL* and *STRA8* were upregulated at 28 and 35 dpf and in the adult phase [25]. *STRA8* is expressed in premeiotic germ cells and induces their meiotic entry. In mice, *STRA8* gene expression was increased in the ovaries of young females paired with young males or elderly male mice, and the level of gene expression was age-dependent [94].

In newborn porcine with 2 days of life, the qPCR analyses revealed that up until 7 days of life, they did not have *OCT4* and *VASA* transcripts [55]. Furthermore, according to the authors, the *OCT4* transcript was only detected after 10–20 weeks in differentiated germ cells (spermatocytes and spermatids). Unlike what was observed in porcine by Goel et al. [54], in this study, the *VASA* transcript was detected in porcine evaluated at 3 days of age. However, OCT4 protein was not found in adult testes; although OCT4 was downregulated in porcine, it was also observed in adult mice testes [95].

Recently, a postnatal mouse ovary study demonstrated the expression of STRA8 in primordial follicles using immunofluorescence and RT-qPCR techniques [96], as observed in porcine ovaries. In adult mouse ovaries, transcripts of *OCT4*, *VASA*, *STELLA*, and *STRA8* were detected, and the same gene transcripts were quantified in oocytes and oogonial stem cells (OSCs), except for STRA8, which was not detected [95]. Another study reports the quantification of *STRA8* transcripts during the female’s reproductive cycle, showing overregulation in the estrus stage [97]. In mice, the gene transcript level can be affected by age, the ovarian follicle reserve, and hormonal activity, which is increased during adulthood, promoting the upregulation of this gene [91]. In female mice, a decrease in OCT4 expression in the adult ovary was observed; this may be related to a decrease in the number of PGCs that have potential pluripotency and an increase in atretic and antral follicles [90,91].

## 5. Conclusions

The present study showed clear morphological differences in the gonads and germ cells after 30 days of gestation, when sexual differentiation occurs in porcine. The formation of the genital ridge can be identified as a protuberance of the mesonephros at 24 days of gestation and continues to develop until 29 days, when differentiation into primitive gonads begins, ending sexual dimorphism at 35 days of gestation. In this period, it was possible to observe apparent histological differences between male and female embryos: in female embryos, at 26 dpf, OCT4 and VASA markers were observed, while at 29 dpf, many cells presented OCT4, VASA, and STELLA markers. STELLA is a maternal protein expressed in PGCs and oocytes. Gene transcription analysis revealed that, regardless of sex, there is a difference in the transcript abundance of the genes OCT4, SOX2, NANOG, STELLA, VASA, PRDM14, and DAZL when different gestational periods or adult tissues are analyzed. The difference in expression could be related to the developmental phase of germ cells in the gonads (early, migration, or late). In the adult phase, the cells of the spermatogenic lineage presented the markers for VASA, DAZL, STRA8, PLZF, and c-Kit, as observed in other species, and the ovaries and markers NANOG, STELLA, VASA, and DAZL were detected, mainly in the cytoplasm of oocytes and primordial follicles, as confirmed by qPCR analysis and described in other species, such as mice. STELLA is important for development and oocyte competence [13]. OCT4, VASA, and DAZL were found to play an essential role in germ cell maturation in mice. In humans, these markers are associated with the development of PGCs and the formation of oocytes and spermatogonia [40]. 

Furthermore, although most migration patterns of PGCs in porcine are similar to those described in other species, mainly in comparison to humans, some differences in embryonic development highlight the need for further studies on the peculiarities of each species. Therefore, the results of this study may help in an unprecedented way in understanding the molecular and epigenetic profile of the porcine germ line from embryonic development to adulthood in both genders, even in future studies of in vitro gametogenesis, which can be used in reproductive biotechnology and as a model for applied studies.

## Figures and Tables

**Figure 1 animals-13-02520-f001:**
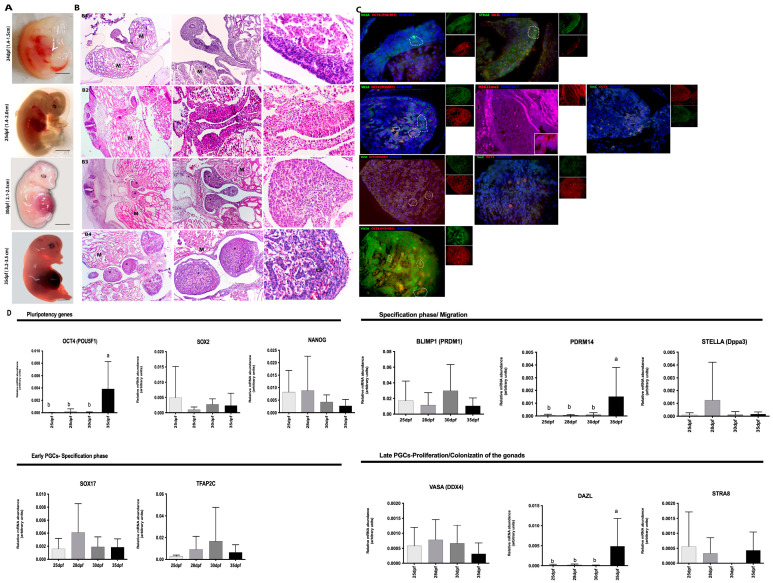
Analysis of gonads from male embryos 24 to 35 dpf. (**A**) Porcine embryos collected on days 24, 25/26, 30- and 35 dpf; (**B**,**C**) (**B1**) at 24 dpf the embryo had an elongated and thickened region on the medial wall of the mesonephric duct (M), with large cells, rounded in shape, with prominent nucleolus, positive for OCT4 (red), VASA (green) and STRA8 (green); (**B2**) At 25 dpf where a small prominence originates towards the body cavity (*). There was no clear tissue organization, as well as a well-defined border. (**B3**) At 30 dpf the mesonephros (M) in prominent tubules located laterally of the future urogenital system and laterally to it the gonadal crests (genital crest) (*). The cells were positive OCT4 and VASA. (**B4**) At 35 dpf, the gonads detach from the mesonephros, and it is already possible to notice the presence of testicular cords (CR), composed of centrally located primordial germ cells and presumably peripherally located supporting cells or Sertoli cells. The germ cells located in the testicular cords (CR), were positive for OCT4 and VASA. (**C**) Analysis of epigenetic markers (H3k9me2 and H3k27me3) were negative in embryos 24, 25/26, 30 and 35 dpf and H3k27me3 (red) was positive in some cells at 26 days of gestation. In addition, pPGCs were positive for 5 mC (green) and OCT4 in embryos at 26 and 30 dpf and (**D**) RT-qPCR analysis showed statistical difference (a,b) the expression of *OCT4, PRDM14*, and *DAZL* was increased in 35 dpf, regardless of sex (*p* < 0.05) (Bar: 50–200 µm and 1 cm).

**Figure 2 animals-13-02520-f002:**
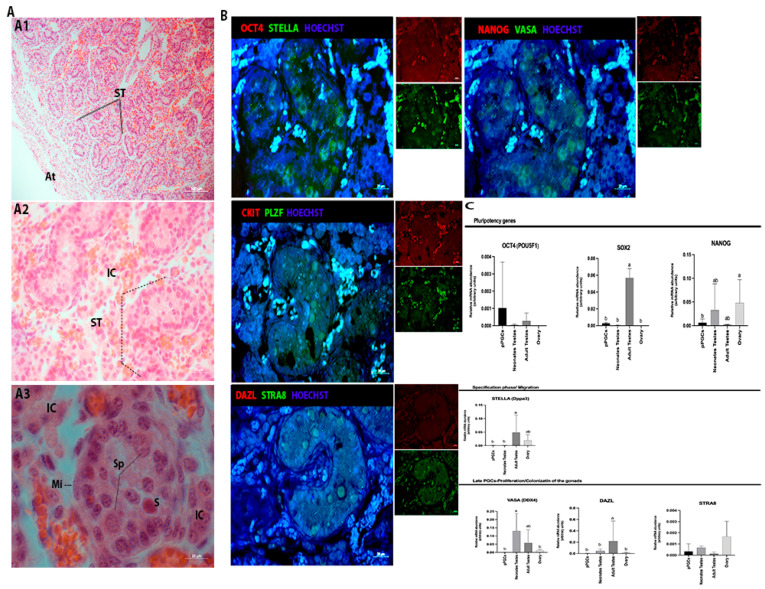
Histological and immunofluorescence analysis of gonads from porcine neonates. (**A1,A2**) Seminiferous tubules (ST) surrounded by blood vessels, interstitial cells (IC) and tunica albuginea (At); (**A3**) Seminiferous tubule (ST) with incomplete spermatogenic process, presence of spermatogonia (Sp). It is also possible to visualize the Sertoli cells (S) close to the basement membrane and myoids cells (MI) and (**B**) At this age, there were positive spermatogonia for PLZF, STRA8, VASA and STELLA (green) and germ cells were negative for OCT4, NANOG and DAZL (red) and (**C**) RT-qPCR analysis showing the gene expression profile of pPGCs, neonates and adults. The statistical analysis showed difference (a, b) of the expression the *SOX2, NANOG, STELLA, VASA* and *DAZL* (*p* < 0.05) (Bar: 20 and 100 µm).

**Figure 3 animals-13-02520-f003:**
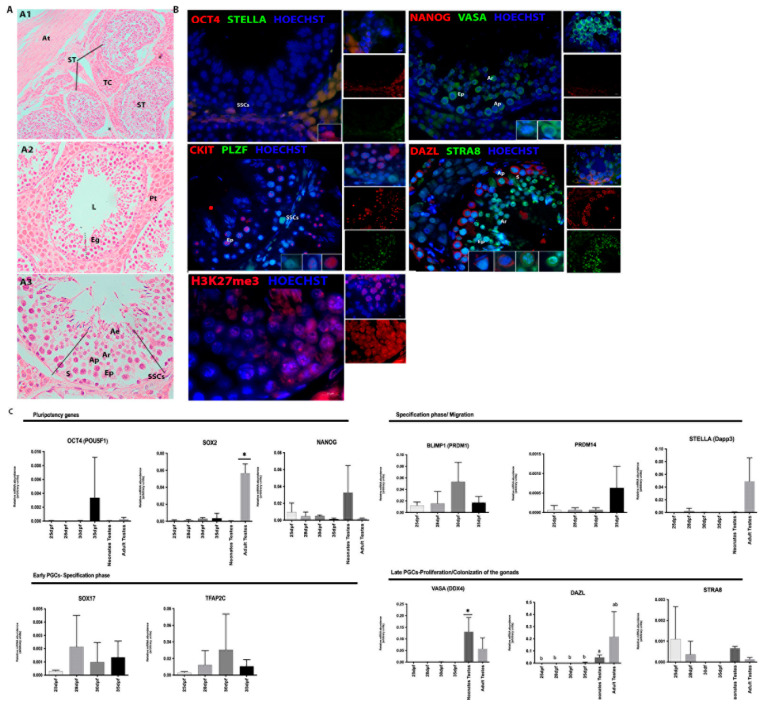
Histological and Immunofluorescence analysis of gonads from porcine adults. (**A**,**B**) Histological and immunofluorescence analysis of adult porcine testes. (**A1**) Testes composed of the Seminiferous tubules (ST) and albuginea tunic (At); (**A2**) The seminiferous tubules were composed of lumen (L) and seminiferous epithelium (Eg) surrounded by interstitial tissue and testicular parenchyma (Pt); (**A3**) Complete spermatogenesis with: spermatogonia (Ap), spermatocytes (Ep), round spermatids (Ar) and elongated spermatids (Ae) spermatozoa and finally, close to the basement membrane, the Sertoli cells (S); (**B**) Immunolocalization of proteins, PLZF (green), DAZL (red), STRA8 (green), VASA (green) and OCT4 (red) in germ cells located in the seminiferous tubules during the spermatogenic process in porcine. PLZF and OCT4 was detected in undifferentiated spermatogonia (SSCs) located close to the basement membrane. DAZL (red) and STRA8 (green) proteins were detected in differentiated spermatogonia (Ap) located near the basal membrane of the tubules and spermatids (Ar), respectively. Co-immunolocalization of STRA8 and DAZL, and some primary spermatocytes (Ep). VASA was detected in some spermatogonia (Ap) and primary spermatocytes (Ep) and round spermatids (Ar). All VASA-positive germ cells in the testes were negative for OCT4 and (**C**) RT-qPCR analysis showing the gene expression profile of pPGCs between 25 and 35 dpf, neonates and adults. The statistical analysis performed difference (a, b and *) on *NANOG* and *DAZL* (*p* < 0.05) (Bar: 50 and 200 µm).

**Figure 4 animals-13-02520-f004:**
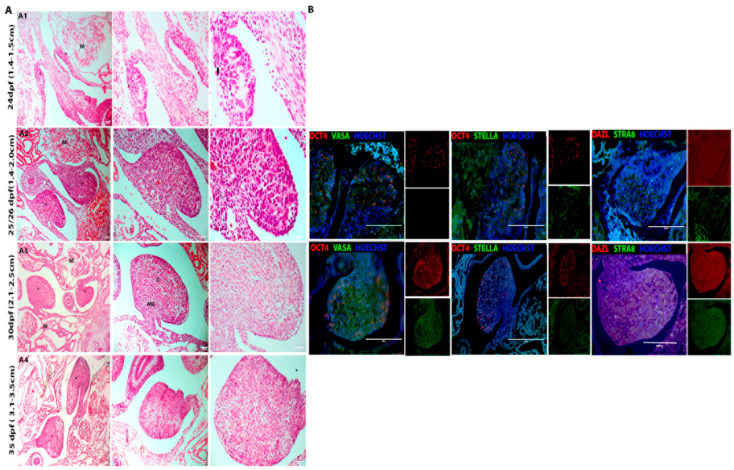
(**A**,**B**) Histological and immunophenotype the pPGCs in female porcine embryos. (**A1**) At day 24 dpf the genital ridges (gonads) (*) were prominences of the mesonephros (M); (**A2**) At 25/26 dpf the gonads (*) were quite prominent, protruding towards the abdominal cavity and with a more rounded shape, however, still connected to the inner wall of the mesonephric duct. In this period, PGCS appeared as large, round cells with prominent nucleoli and were positive for OCT4 (red) and VASA (green); (**A3**) At 30 dpf the primitive gonads (*) were practically independent of the mesonephros (M). Many cells were OCT4 positive. Furthermore, some cells were VASA, STELLA and STRA8 (green) positive and (**A4**) At 35 dpf the mesonephros independent gonads (*) with primordial germ cells surrounded by follicular cells (flat somatic cells derived from the surface epithelium of the developing ovary) (Bar: 50 and 200 µm).

**Figure 5 animals-13-02520-f005:**
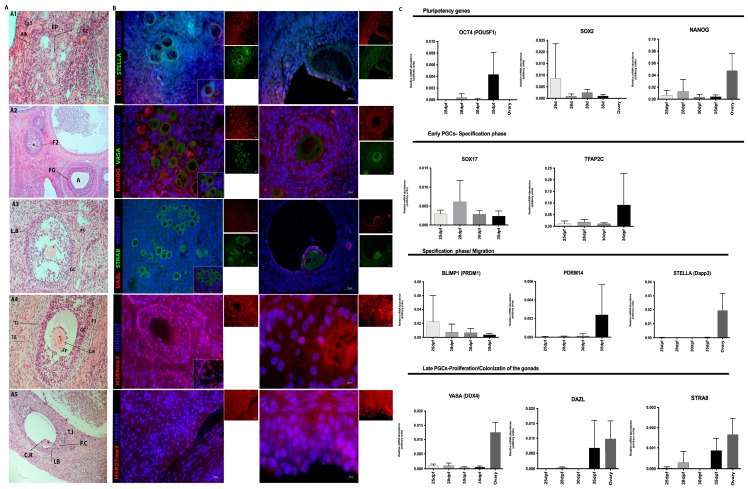
Histological and immunofluorescence analysis of porcine ovaries. (**A1**) Primordial follicles (FP) composed of primary oocyte surrounded by simple squamous epithelium of follicular cells (F.C), epithelium superficial (Ep.S) and albuginea Tunic (Alb); (**A2**) Secondary follicle (F2) composed of primary oocyte (*) surrounded by a glycoprotein layer, the zona pellucida, and a stratified epithelium of granulosa cells (GC). Follicle the Graaf (FG) with evident antrum (A); organized granulosa cells and a thick layer of theca organized as externa and interna; (**A3**) Tertiary follicle (F3) with the beginning of follicular cavity development and granulosa cell (GC) stratification and basal lamina (L.B); (**A4**) Tertiary follicle (F3) whose primary oocyte (*) is surrounded by a layer of *cumulus oophorus* (C.R) in the process of organization; stratified epithelium of granulosa cells (GC); cells of theca interna (T.I) and externa (T.E); (**A5**) Tertiary follicle (F3) containing a very evident follicular cavity (antrum); single-layered *cumulus oophorus* (C.R) cells surrounding the primary oocyte (*); stratum granulosum; layer of connective tissue, the theca, divided into interna (T.I) and externa (T.E); (**B**) Immunofluorescence analysis of adult porcine ovaries, with the presence of positive cells for NANOG (red), VASA, STELLA, STRA8 (green) and DAZL (red) markers. In addition, in adult ovaries, H3k9me2-positive (red) follicular cells were detected in the primary follicle and H3k27me3 (red) in interstitial cells and in the cells that make up Graff’s theca follicle. (**C**) The RT-qPCR analysis showed that pPGCs between 25–35 dpf and ovaries show alteration in the expression of all analyzed genes, related to the processes that these cells participate during each stage of development, although there is no statistical difference (Bar: 10, 20, 50 and 100µm).

**Figure 6 animals-13-02520-f006:**
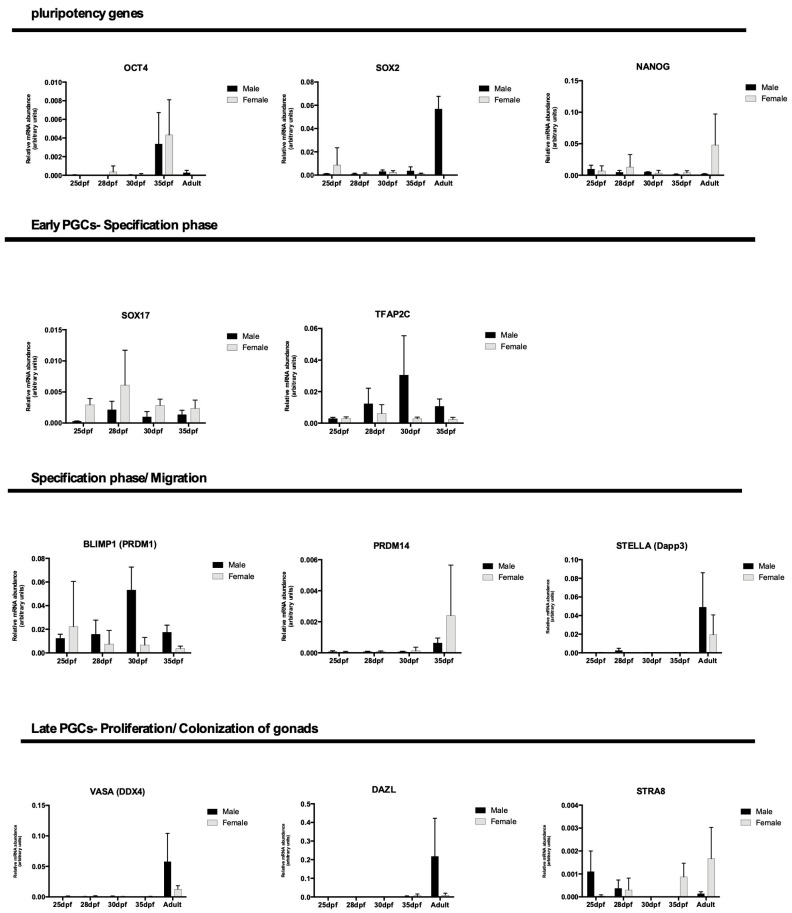
RT-qPCR analysis showed that pPGCs between 25- and 35 dpf and adults show changes in the expression of all genes analyzed, related to the processes that these cells participate in during each stage of development. The SOX17 expression was different between genders (*p* = 0.0076), regardless of gestational age.

**Table 1 animals-13-02520-t001:** Sequence of the porcine primers.

Sequence Name	Sequence	Size (bp)	Gene Bank
*OCT4* *(POU5F1)*	GGGTTCTCTTTGGGAAGGTGT CTCCAGGTTGCCTCTCACTC	224	NM_001113060.1
*SOX2*	CTCAGTGGTCAAGTCCGAGG AGAGAGAGGCAGTGTACCGT	223	NM_001123197.1
*NANOG*	TCCTTCCTCCATGGATCTGCT GGGTCTGCGAGAACACAGTT	155	NM_001129971.1
*SOX17*	TCCACTCTGCTAGTGCCTCT CTGGGGATGCCCTAATGTTCA	193	XM_001928376.7
*TFAP2C*	GAAACCCTGGACTGGACGAG GTAGCACCACTTGCAGAGGA	121	NM_001123201
*PRDM14*	AGTGGATGCTTCTCTGCTACC TGCCTTTCTCTCTTGGTTCA	112	[26]
*BLIMP1* *(PRDM1)*	CAGTGCCGTGAAGTTTCC AAGGATGCCTCTGCCTGAAC	186	[26]
*VASA (DDX4)*	CCTGCCCAGGAATGCCATCA ACTGGCCAACTTGGAGAATGGT	180	[26]
*STELLA* *(Dapp3)*	CCCGCCTTTCAATCTGTCTCC TCGCCGAACCGTGTATCGAA	219	[26]
*DAZL*	GGTCGCTTTGCTTATCCGC TGCAGCAGACATTACTGCGA	158	[26]
*STRA8*	TGGAGAAGGGAGCAACCCCA ACCTGCCACTTTGAGGCTGT	189	[26]
*β-ACTIN*	GAAGATCAAGATCATCGCGCCT GTGGAATGCAACTAACAGTCCG	117	XM_003124280.4
*GAPDH*	GTCGGTTGTGGATCTGACCT ACCAGGAAATGAGCTTGACGA	221	NM_001206359.1

## Data Availability

The data used to support of this study are available from the corresponding author on reasonable request.

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
