# Peer review of "Porcine Germ Cells Phenotype during Embryonic and Adult Development"

_animals, 2023, doi:10.3390/ani13152520_

Round 1

Reviewer 1 Report

This manuscript is informative and impressive. The results are significant to germ cells field because they exhibit the progression of a diversity of markers during pPGCs, and they may allow use in applied science as reproductive and regenerative medicine.

I think this manuscript should be published. Only a few mistyping should be corrected.

Reviewer 2 Report

This paper entitled “Procine germ cells phenotype during embryonic and adult development” described a study about the dynamics of pluripotent, germline, and epigenetic markers of procine primordial germ cells through gestational and adult stages. They performed their studies in both male and females, examining the expression of different markers that are associated with PGC development and differentiation in porcine. Immunostaining and RT-qPCR were applied to evaluate gonads at different developmental stages. Their results demonstrate different morphological patterns between different sexes, as well as distinct transcript features throughout the developmental stages.

Overall, this paper presents findings that could contribute to understanding the pathways of PGCs during reproductive development. Understanding the mechanisms of primordial germ cells is important for the formation and reproduction, making this study is meaningful to the field of reproductive biology. I have the following questions/comments prior to its publication.

1.       All the figures have low resolution, making it impossible to zoom into the details, I suggest to the authors to increase the quality of their figures and transfer them into vector images.

2.       Please explain the reason for choosing these markers (e.g. SOX17 OCT4) for the study and the general functions of them in the introduction session.

3.       Line190-197: please indicate the figure panels for each statement.

4.       Line213: delete “Bar”

5.       Line 288: I could not see signals in VASA.

6.       Line 367: please specify the P value is for age difference?

Overall, English is okay, but some sentences are quite wordy, making it difficult to understand.

Reviewer 3 Report

The present research explains "the expression of epigenetic markers and the different pluripotent and germline markers associated with the development and differentiation of porcine PGCs (pPGCs), with the aim of understanding the different gene expression profiles between sexes". This research provides interesting information. However, some important changes need to be made before final publication. 

Abstract: review the "Journal" guidelines. It is mentioned in "MDPI Style Guide" the following: "The abstract contains a summary of the entire paper and can be up to 200 words long with only one paragraph". (https://www.mdpi.com/authors/layout) In this case it exceeds the number of words. Therefore, this section should be restructured.

Line 40.- if this section is defining abbreviations, why not define "dpf" as "days post-fertilization"?

INTRODUCTION

Line 55.- they mention "Kobayashi et al in 2017," is this the correct way to cite? Because in line 85 they mention "Kobayashi and colleagues [4]", I recommend homogenizing citations.

Line 96.- change the word "in vivo" by “in vivo”.

MATERIAL AND METHODS

General comments: I recommend mentioning which were the selection criteria for these animals, body condition, etc. Also, if a clinical study was performed before the study to rule out any clinical pathology. I recommend mentioning the number of samples evaluated for each animal. Also, only females and embryos are mentioned, but in the results section they mention "analysis of gonads from porcine adults." Nothing is described about them.

Line 175.- in the section "Statistical analysis" the authors mention that the data came from a normal population, otherwise they were transformed. However, in order to perform the ANOVA test it is necessary that the data have homoscedasticity of variances, what test did they perform to prove this? Also, you mention "the significance level considered was 5%" as significance level, I recommend you to mention (P<0.05).

RESULTS

General comments:

I recommend restructuring some figures, some graphs are too small. The following is mentioned.

Line 200- Figure 1. Section "D" graphs and literals are too small to understand. I recommend to increase the size. Line 203, add space "(Green); B2)" and here also "25dpf".

Line 212-213.- mention "and adults present statistical differences in the expression of OCT4, SOX2, NANOG, STELLA (Dapp3), VASA (DDX4), PRDM14 and DAZL related to the processes in which these cells participate during each stage of development (p>0.05)".  Therefore, it would be "(p<0.05)". Also, they mention "Bar: Bar:", check this.

Line 248.- Figure 3. In section "C" the graphs and literals are too small to understand. I recommend increasing the size.

Line 260.- they mention "35 days post-fertilization" and in other sections they mention "dpf", I recommend to homogenize this in the whole document. As well as to homogenize its use, this due to the fact that in line 37 and 109 they mention "days post". While in lines 201 and 260 they mention "days pos".

Line 304.- Figure 5. Section "c" graphs and literals are too small to understand. I recommend increasing the size.

Line 311.- change "external (T.E) A5)" to "external (T.E); A5)".

Line 313.- separate "(T.I)and external".

DISCUSSION

In general, I recommend restructuring this section and explaining the findings observed in the research and trying to explain the mechanism involved. In addition, it is advisable to do this section according to the order in which the results were mentioned. That is, we start with a "Histological and immunophenotype" analysis of the gonads of males and females. Subsequently, reference is made to "Analysis of the gene expression in porcine germ cells" and it is mentioned "The expression of the SOX17 gene showed a statistical difference related to sex, regardless of gestational age (p>0.05):" Add information on what may be the cause of this elevation in females in line 405.

In addition to how the markers were found with "OCT4, NANOG, STELLA, VASA, c-Kit, STRA8, DAZL and PLZF," in the different embryonic developments. It is very important to explain "the expression of epigenetic markers and the various pluripotent and germline markers associated with the development and differentiation of PGCs in porcine (pPGCs)" in both sexes. That is the aim of this study.

Line 372.- "PGC differentiation process" is mentioned, however, you have used the term "PGCs". Check this.

Line 380.- you mention "10 dpc," do you mean "days post-conception"?

Line 383.- mention "Parma and colleagues (1999)[15]", please correct this.

Line 389.- separate "30dpf".

Line 392.- mention "Li (2000)[34]", please correct it.

Line 436.- change "similar" to "Similar".

Line 458.- mention "Pieri et al. (2022)[18]", please correct it.

Line 457-458.- mention "similar to that 457 described by Pieri et al. (2022)[18] As described by the authors,". Isn't this somewhat redundant?

Line 472.- they mention "According to [49,50],". I recommend mentioning it as an active citation.

Line 539.- change "In vitro" to "In vitro" in italics.

Line 539.- mention "according to [18]". I recommend mentioning it as an active citation.

Line 541.- change "in vivo" to "in vivo" in italics.

Line 544-545.-mention "An and colleagues (2015) [66]", please correct it.

CONCLUSION

I recommend restructuring this section and specifying the findings mentioned in the results.

Reviewer 4 Report

The authors have thoroughly investigated the developmental dynamics of porcine Primordial Germ Cells (pPGCs), utilizing methods like immunofluorescence and RT-qPCR. Identifying sex-specific differences in pPGCs across embryonic and adult stages offers valuable insights into mammalian reproductive biology and sexual differentiation.

Although my expertise does not reside within this field, I have focused on methodology, data representation, and statistical analysis, and I am pleased to report that these aspects of the manuscript appear sound and well-executed.

One minor suggestion I would make is to add subtitles in the discussion section. This could help enhance the clarity and improve the readability of this section. Overall, the authors have produced a well-structured and thoughtfully composed paper.
